# Significant relaxation of SARS-CoV-2-targeted non-pharmaceutical interventions may result in profound mortality: A New York state modelling study

Benjamin U. Hoffman [ID] [1,2] *

1 Vagelos College of Physicians and Surgeons, Columbia University, New York City, New York, United States of America, 2 Department of Medicine, University of California San Francisco, San Francisco, California, United States of America

* Benjamin.Hoffman@ucsf.edu

**Data Availability Statement:** The custom-built software used generate this model, along will scripts to create the analyses and data within this

## Abstract

Severe acute respiratory syndrome-coronavirus 2 (SARS-CoV-2) is the most significant global health crisis of the 21st century. The aim of this study was to develop a model to simulate the effect of undocumented infections, seasonal infectivity, immunity, and non-pharmaceutical interventions (NPIs) on the transmission, morbidity, and mortality of SARS-CoV-2 in New York State (NYS) based on data collected between March 4 and April 28, 2020. Simulations predict that undocumented infections significantly contribute to infectivity, NPIs are effective in reducing morbidity and mortality, and relaxation >50% of NPIs from initial lockdown levels may result in tens-of-thousands more deaths. Endemic infection is likely to occur in the absence of sustained immunity. As a result, until an effective vaccine or other effective pharmaceutical intervention is developed, the risks of significantly reducing NPIs should be carefully considered. This study employs modelling to simulate fundamental characteristics of SARS-CoV-2 transmission, which can help policymakers navigate combating this virus in the coming years.

## Introduction

The global pandemic of severe acute respiratory syndrome-coronavirus 2 (SARS-CoV-2) has emerged as the most significant global health crisis of the 21st century. Since SARS-CoV-2 surfaced in the city of Wuhan, Hubei, China in December of 2019, the virus has caused significant global morbidity and mortality with over 3 million cases and 235,000 deaths worldwide [1–3]. By late April 2020, the United States had accumulated the most cases of SARS-CoV-2 globally, with New York State (NYS) emerging as the epicenter [3]. Local and federal governments have attempted to restrict the spread of SARS-CoV-2 through mandating social distancing, suspending non-essential services, and increasing the testing capacity of health systems [4]. Key unanswered questions are: How effective are these measures at diminishing the active pandemic, and how long might they need to last?

manuscript, is freely available at https://github.com/buh2003/BUHoffman_COVID.

**Funding:** BUH was funded by a NIH MSTP training grant (National Institute of General Medical Sciences T32GM007367; https://www.nigms.nih.gov). The funders had no role in study design, data collection and analysis, decision to publish, or preparation of the manuscript.

**Competing interests:** The authors have declared that no competing interests exist.

To answer these key questions, a compartment model was employed to simulate the transmission dynamics of SARS-CoV-2. This model is capable of simulating the effects of non-pharmaceutical interventions (NPIs), as well as essential host-pathogen characteristics such as undocumented infection, immunity, and temperature- and humidity-dependent infectivity. Undocumented infections were defined as individuals infected with SARS-CoV-2, but either due to failure of testing or having been asymptomatic, never undergo quarantine and continue to infect others. The implementation of these factors in a dynamic model of SARS-CoV-2 is essential to not only understanding the fundamental epidemiological dynamics of the current SARS-CoV-2 outbreak, but also to shed light what might drive endemic infection [5, 6].

Within the epidemiological conditions explored in this study, simulations indicate that SARS-CoV-2 is a highly infectious pathogen, whose transmission dynamics may be driven by large numbers of undocumented infections. Simulations of relaxed NPIs during the summer of 2020 in NYS project that modest relaxation may have minimal effects, but reduction >50% may significantly increase transmission and mortality. Projection through 2021 predicts a second outbreak of SARS-CoV-2 in NYS during the winter months, with possibly tens-of-thousands of additional deaths if NPIs are not resumed. Lastly, simulation of immunity parameters indicates that durable sustained immunity to SARS-CoV-2 might be required to prevent establishment of this virus as an endemic pathogen. Together, these simulations explore key characteristics of SARS-CoV-2 transmission dynamic. The findings of this study further our understanding of SARS-CoV-2, and contribute to the ongoing global effort to combat this virus.

## Results

### Simulating the effect of undocumented infections on SARS-CoV-2 transmission

Undocumented infections represent an increasingly important factor in the spread of SARS-CoV-2. The number of undocumented, but infectious, individuals is a critical variable in modulating the infectivity of respiratory viruses [6, 7]. Recent studies suggest that between 40–95% of all SARS-CoV-2 cases may be undocumented [8, 9]. Serological data from NYS indicate that ~10–20% of the affected population have detectable antibodies to SARS-CoV-2, even though only ~2% had tested positive by late April, 2020 [10]. Based on these studies, an undocumented infection rate of 75% was selected for this modelling work. An 11-compartment model was developed to comprehensively model SARS-CoV-2 transmission dynamics (see Methods and S1 Appendix; S1 Fig). The model was optimized to SARS-CoV-2 case data from New York State collected between March 4th and April 28th, 2020 with the particle swarm algorithm, and confidence intervals were estimated by introducing lognormal gaussian noise to the source data and randomly sampling initial parameter estimates (see Methods; S1 and S2 Tables) [11, 12]. Sensitivity analysis revealed the model was most sensitive to variations of β, which represents the average number of contacts per individual per day (see Methods; S2 and S3 Figs). Comparison of the model to NYS case-data revealed a good fit (see Methods; S4 Fig).

This model represents an important advancement over recent studies as it simulates the effects of undocumented infections, seasonal variability, and sustained immunity on SARS-CoV-2 transmission dynamics in NYS (Fig 1A–1C; S5 and S6 Figs) [5, 13–17]. The seasonal variability of SARS-CoV-2 transmission was modeled based on published non-SARS-CoV-2 human coronavirus (HCoV) testing data (see Methods). Based on these conditions, projecting through September 1st, 2020, if the NPIs implemented on March 22nd, 2020 are not relaxed, the SARS-CoV-2 outbreak will likely be in decline by mid-July 2020, with a plateau in cumulative cases: 1.53 million people in NYS will be infected (undocumented infected+ confirmed

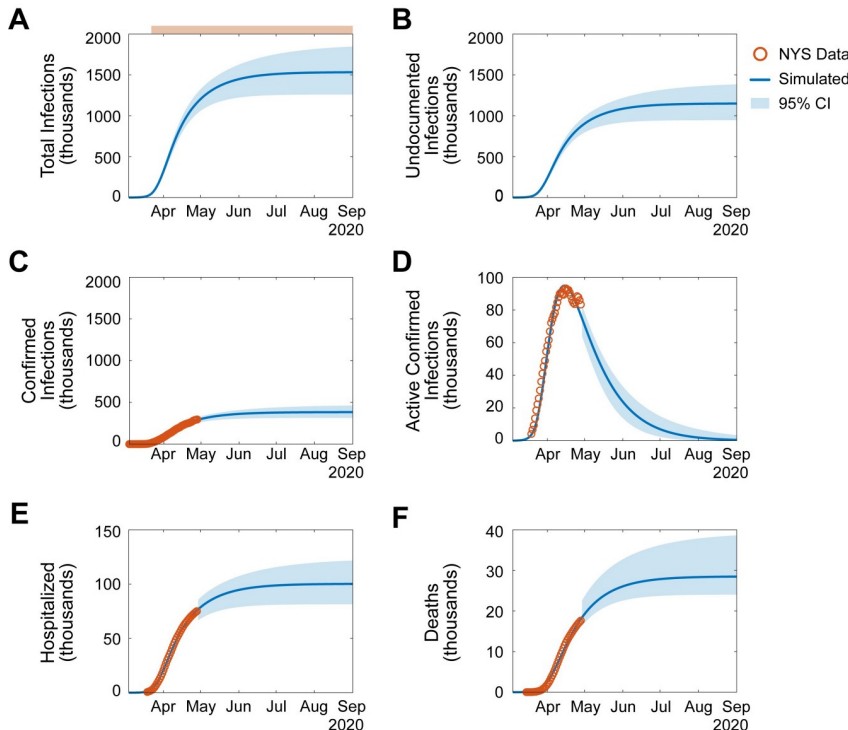

**Fig 1. Simulating the effect of undocumented infections on SARS-CoV-2 transmission.** (A-F). Simulation of SARS-CoV-2 transmission dynamics through September 1, 2020. NPIs signified as in (A) top: pink, NPIs initiated on March 22$^{nd}$, 2020. Orange circles, NYS SARS-CoV-2 data (see Methods). Blue line, simulated projection. Light blue box, 95% confidence interval. **A.** Cumulative total infections (undocumented infections + confirmed symptomatic infections). **B.** Cumulative undocumented infections. **C.** Cumulative confirmed symptomatic infections. **D.** Active confirmed infections. **E.** Cumulative hospitalizations. **F.** Cumulative deaths. See also, S2–S6 Figs, S3 Table.

symptomatic infected; 1.26–1.79 million, 95% CI), with 100,000 total hospitalizations (80,000–120,000, 95% CI), and 28,500 deaths (22,000–37,600, 95% CI; Fig 1D–1F; S3 Table) [18]. The maximum basic reproductive number ($R_0$), which describes the infectivity of a pathogen, over this period was 5.7 (5.3–6.0, 95% CI), significantly greater than the $R_0$ reported by other published models of SARS-CoV-2 ($R_0$ = 1.6–3.0) [13, 14, 19]. To account for the rapid growth in testing availability and implementation over the initial period of the outbreak, the maximum basic reproductive number was estimated after the daily rate of change in the ratio of positive tests to confirmed tests was negative for 7 consecutive days (see S1 Appendix). These simulations suggest that the infectious potential of SARS-CoV-2 has thus far been underestimated, and that undocumented infections may be primary drivers of transmission.

## Reduction of NPIs by >50% may result increased transmission and significant mortality

Next, the short-term effects of relaxed NPIs in NYS were examined. To do so, a rapid decline in NPIs to a steady-state level over time was simulated until September 1$^{st}$, 2020 (Fig 2A–2C; S7 Fig). Moderate reduction of NPIs (≤50%), independent of the time at which the relaxation was initiated, resulted in minimal differences in total deaths by September 1$^{st}$, 2020 (total deaths: 30%, 29,600 [22,600–40,900, 95% CI]; 50%, 32,200 [23,200–49,200, 95% CI]; Fig 2D; S4 Table). Next, the instantaneous effective reproductive number ($R(t)$) was analyzed to quantify the magnitude of NPI reduction required to initiate a second outbreak of SARS-CoV-2. $R(t)$

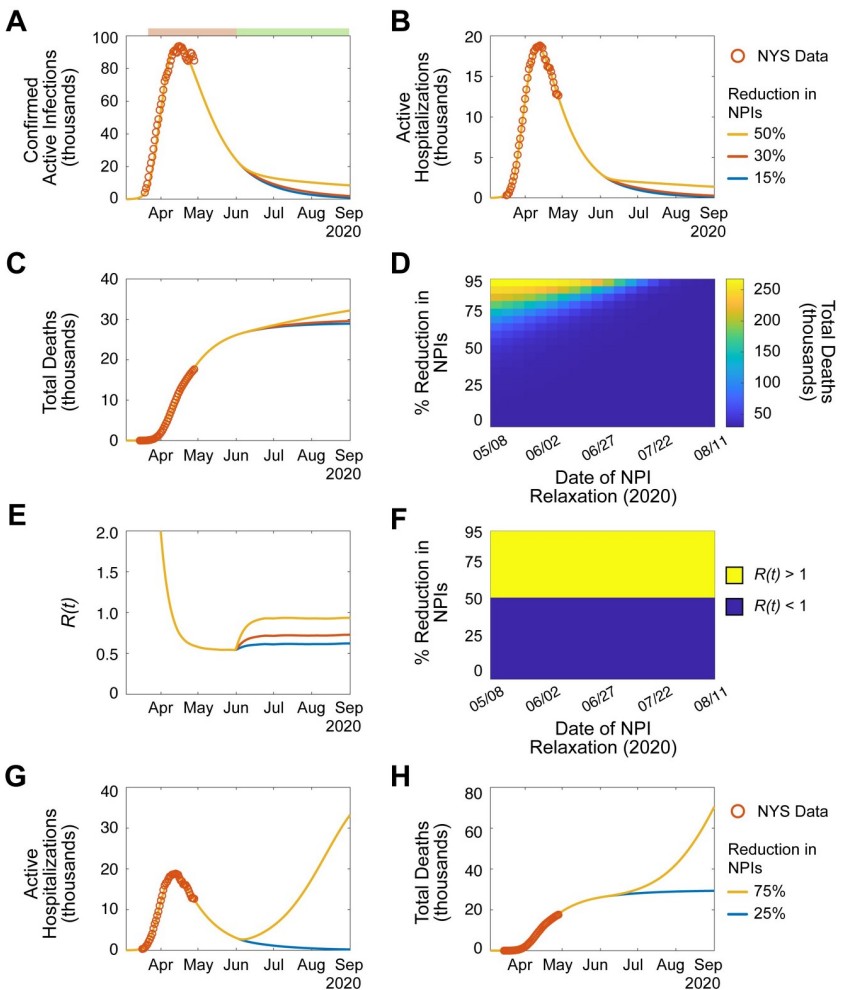

**Fig 2. Reduction of NPIs by >50% may result increased transmission and significant mortality. (A-F).** Simulation of SARS-CoV-2 transmission dynamics in the presence of NPIs through September 1, 2020. Periods of NPIs signified as in (A) top: pink, increased NPIs; green, relaxed NPIs. Orange circles, NYS SARS-CoV-2 data. Lines, simulated projection of reduced NPIs starting June 1, 2020 (yellow, 50% reduction; red, 30% reduction; blue, 15% reduction). **A.** Active confirmed infections. **B.** Active hospitalizations. **C.** Cumulative deaths. **D.** Heatmap displaying the effect of NPI magnitude and date of reduction on the number of cumulative deaths. **E.** *R(t)*. **F.** Categorical heatmap displaying the effect of NPI magnitude and date of reduction on *R(t)* > 1 (yellow, *R(t)* > 1; blue *R(t)* < 1). **(G-H).** Simulation of extreme reduction of NPIs on June 1, 2020 (yellow, 75%; blue, 25%). **G.** Active hospitalizations. **H.** Cumulative deaths. See also, S7 and S8 Figs, S4 Table.

differs from $R_0$ as it describes the fraction of the population that is susceptible to infection as a function of time [20]. Analysis of *R(t)* enables temporal estimation of the potential for an outbreak in a population where not all individuals are susceptible: where *R(t)* < 1 results in decreasing infections, and *R(t)* > 1 results in increasing infections. Only simulations where NPIs were relaxed >50% resulted in *R(t)* > 1 at any time before September 1[st], 2020 (Fig 2E and 2F) [21]. Simulation of reductions > 50% resulted in the rapid development of a second wave of SARS-CoV-2 cases, with a dramatic increase in deaths by September 1[st], 2020 (25% vs 75% reduction: 29,300 [22,500–40,000, 95% CI] vs. 70,500 [31,200–97,900, 95% CI] deaths; Fig 2G and 2H; S4 Table). Based on the conditions of these simulations, these data indicate that reducing NPIs by up to 50% in NYS may not significantly increase transmission before

September 1st, 2020. However, reduction greater than 50% will likely result in significant mortality.

Multiple organizations, including the White House, the Centers for Disease Control and Prevention (CDC), and the NYS Department of Health (NYSDOH) have proposed a phased model for relaxing NPIs [22, 23]. These models employ a "wait-and-see" approach to lessening restrictions in order to prevent increased transmission of SARS-CoV-2. To test if a phased approach was superior to a one-time relaxation of NPIs, serial NPI relaxations, each with a duration of 14 days, were simulated at various time points (S8 Fig). For the majority of relaxation magnitudes, one-time relaxation was comparable to phased relaxation; however, phased relaxation was superior for large magnitude reductions in NPIs (one-time 75% vs. phased 75%: 70,500 [31,200–97,900, 95% CI] vs 38,200 [23,800–66,700, 95% CI] deaths by September 1st, 2020; S5 Table). Together, these simulations demonstrate that phased relaxation may be useful in the context of significant NPI reduction.

## Recurrent outbreak of SARS-CoV-2 in NYS in early 2021

Given the social and economic toll and magnitude of the current SARS-CoV-2 pandemic, a key question is: will there be a recurrent outbreak in the next 1.5 years? To answer this question, the course of SARS-CoV-2 was simulated through September 1st, 2021 (Fig 3A–3C; S9 Fig). Indeed, given the conditions of the model, a SARS-CoV-2 outbreak is predicted to occur throughout the winter months of 2020/21. Simulations indicate that the magnitude of the outbreak depends on the resumption of NPIs, where stronger measures implemented early result in reduced morbidity and mortality through September 1st, 2021 (% NPIs, cumulative deaths: 100%, 58,000 [28,900–90,800, 95% CI]; 75%, 92,000 [33,600–109,600, 95% CI]; 50%, 154,500 [43,500–183,000, 95% CI]; Fig 3D). However, analysis of the $R(t)$ indicates that the potential for a second outbreak is extremely high, where no combinations of NPI strength and implementation date were successful in preventing $R(t) > 1$ (Fig 3E and 3F). Importantly, the increased winter-time transmission reflects the model's assumption that the seasonal variation in transmission dynamics of SARS-CoV-2 are similar to that of other non-SARS-CoV-2 human coronaviruses. Given this assumption, simulations predict recurrent outbreak in early 2021 that may be mitigated, but not avoided entirely, through the resumption of NPIs.

## Simulation of sustained immunity to SARS-CoV-2 predicts endemic potential

Respiratory viruses, including non-SARS-CoV-2 HCoVs and influenza are endemic across the world, in part due to the seasonal variability of their infectivity and incomplete sustained immunity [24, 25]. Thus, it is essential to explore what factors might drive endemic infection of SARS-CoV-2 in NYS. To do so, the transmission dynamics of SARS-CoV-2 were simulated through September 1st, 2025 (Fig 4A and 4B; S10 Fig). Indeed, given the hypothetical conditions of the simulations, SARS-CoV-2 emerged as an endemic pathogen in NYS despite resumption of NPIs for 3 months in the winter of 2020/21. Next, the effects of differential immune parameters (% infected individuals that become immune, and duration of immunity) on cumulative deaths and endemic potential were analyzed (Fig 4C and 4D). Only parameters conferring large proportions of infected individuals (>70%) with long-term sustained immunity (>15 years) resulted in eventual elimination of annual SARS-CoV-2 infection over 5 years. These simulations demonstrate that the potential for SARS-CoV-2 to become an endemic pathogen in NYS likely depends on the quality of immunity that individuals develop. As a result, the development of an efficacious vaccine will be essential to prevent endemic SARS-CoV-2 infection in NYS.

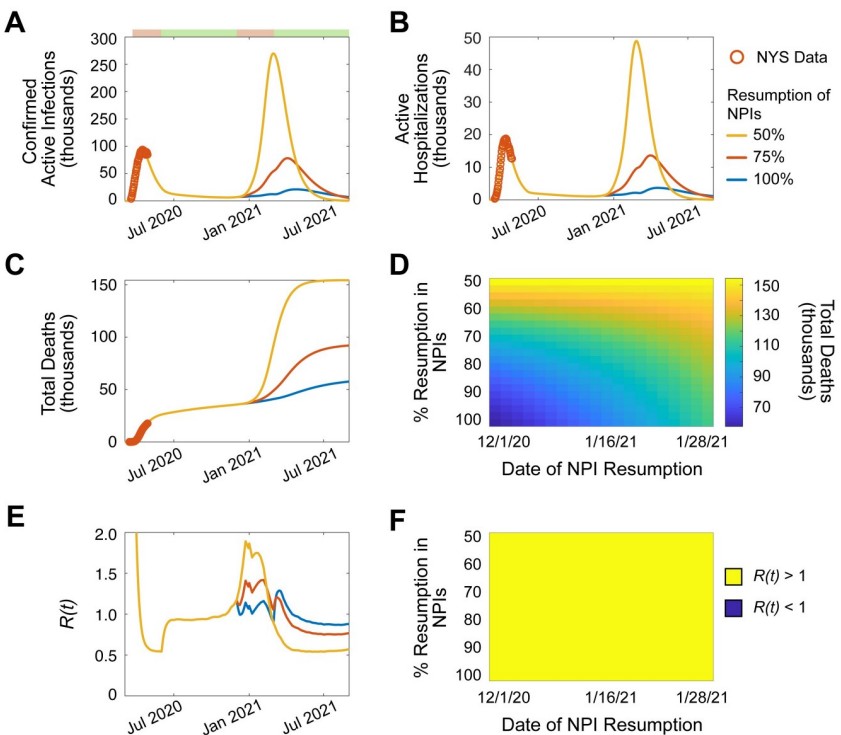

**Fig 3. Recurrent outbreak of SARS-CoV-2 in NYS in early 2021.** (A-F). Simulation of SARS-CoV-2 transmission dynamics in the presence of NPIs through September 1, 2021. Simulation assumes 50% NPI reduction on June 1, 2020. Periods of NPIs signified as in (A) top: pink, increased NPIs; green, relaxed NPIs. Orange circles, NYS SARS-CoV-2 data. Lines, simulated projection with resumption of NPIs on December 1, 2020 to Feb 28, 2021, and relaxation after March 1, 2020. **A.** Active confirmed infections. **B.** Active hospitalizations. **C.** Cumulative deaths. **D.** Heatmap displaying the effect of NPI magnitude and date of resumption on the number of cumulative deaths. **E.** $R(t)$. **F.** Categorical heatmap displaying the effect of NPI magnitude and date of resumption on $R(t) > 1$ (yellow, $R(t) > 1$; blue $R(t) < 1$). See also, S9 Fig.

## Discussion

As the global health community continues to grapple with issues caused by the growing spread of SARS-CoV-2, it is essential to develop insights into the complex transmission dynamics of this pathogen. Mathematical models can be useful to reveal epidemiological characteristics of infections [26]. Here, an epidemiological model was developed to predict the spread, morbidity, and mortality of SARS-CoV-2 in NYS. This model is designed to simulate key features of the virus, such as the effect of undocumented infections on transmission, the development of sustained immunity to the virus, change in infectivity with weather, and the effect of non-pharmaceutical interventions to mitigate the outbreak. This study indicates that dynamic NPIs are likely critical to controlling the spread of SARS-CoV-2, and that in the absence of development of profound immunity caused by infection or the development of an efficacious vaccine, the virus may become endemic.

One important finding of this study is that based on the conditions of the model, simulations indicate that undocumented infections may have fueled the rapid spread of SARS-CoV-2 in NYS. During the recent NYS outbreak, the model estimated a peak $R_0$ of 5.7 (5.3–6.0, 95% CI), reaching a baseline of 4.4 (4.1–4.7, 95% CI) by July 2020, greater than reported in other publications (S6 Fig): which estimated 2.2 in the United States, and 2.2–2.6 in Wuhan and other regions of China [5, 13, 14, 27]. The magnitude of the model's estimated $R_0$ reflects the assumption that a large number of undocumented infections existed in NYS at the start of the

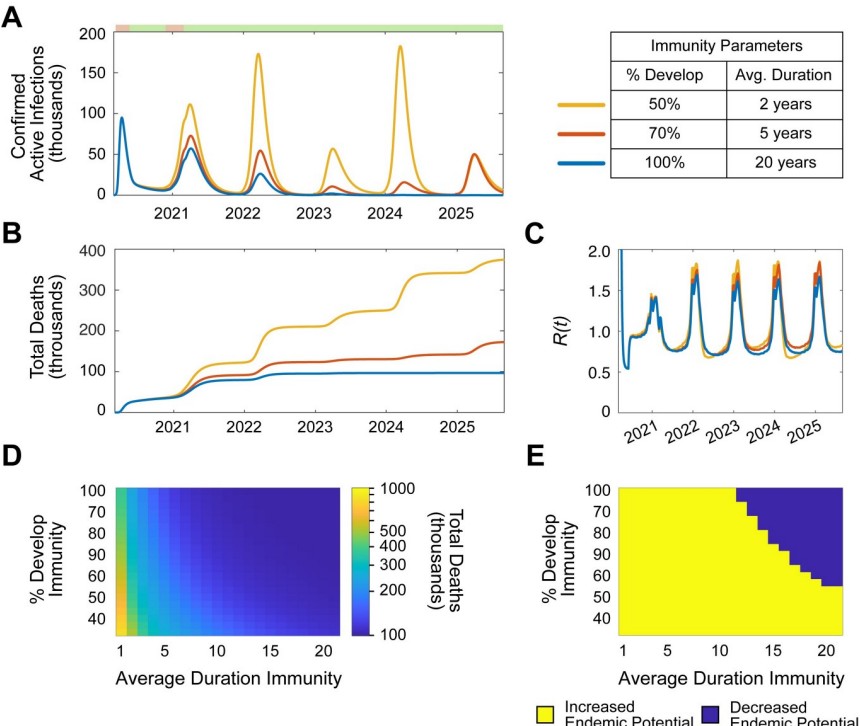

**Fig 4. Simulation of sustained immunity to SARS-CoV-2 predicts endemic potential.** (A-D). Simulation of SARS-CoV-2 transmission dynamics in the presence of NPIs through September 1, 2030. The following NPI parameters were assumed: 50% from June 1 to October 31, 2020; 75% from November 1, 2020 to March 1, 2021; 50% after March 1, 2021. Periods of NPIs signified as in (A) top: pink, increased NPIs; green, relaxed NPIs. (A-B). Simulation varying the % of individuals who develop immunity and the average duration of immunity. **A.** Active confirmed infections. **B.** Cumulative deaths. **C.** Heatmap displaying the effect of immunity parameters on cumulative deaths through September 1, 2025. **D.** Heatmap displaying the effect of immunity parameters on endemic potential. Endemic potential defined as the presence annual recurrence (average total annual infections > 20,000) from September 1, 2023–2025. See also, S10 Fig.

outbreak. Based on serological studies, a 75% undocumented infection rate was implemented in this study (see Methods) [8–10]. As a result, the model projects 868,000 (795,000–947,000, 95% CI) undocumented infections by April 28th, 2020, despite the NYSDOH reporting only 295,106 confirmed infections. It is important to note, however, that estimation of $R_0$ during the initial outbreak of an epidemic is confounded by the concurrent exponential increase in infections and testing. To account for this, estimation of the maximum basic reproductive number was approximated only when the percent positivity of confirmed tests was negative for 7 consecutive days (see S1 Appendix). However, despite this method, it is possible the findings of this study may represent overestimation of infectivity. Together, these simulations demonstrate the significant effect undocumented infections may have had in driving the exponential growth of the outbreak in NYS: likely, undocumented infected individuals were unaware of having contracted infection with SARS-CoV-2, and rather than entering quarantine, continued to expose contacts with the virus.

The simulation of NPIs on modulating SARS-CoV-2 transmission in NYS reveals that NPIs are effective in reducing the spread of the virus; however, to prevent the development of subsequent large magnitude outbreaks, they will likely have to persist in some form for multiple years. This model projects that these measures can be reduced modestly to about 30% of current levels without having a significant impact on SARS-CoV-2- related morbidity and

mortality. However, if they are reduced more than 50%, $R(t)$ will likely increase >1, and mor-bidity and mortality may dramatically grow. NPIs will need to rely not only on social distanc-ing, but also on mask-wearing, improved testing, isolation, and contact tracing. Indeed, intensive testing and case-based interventions have become essential to control efforts in coun-tries across the globe [28–30]. To simulate this long-term steady state of improved NPIs, the NPI variable was never allowed to reach 0% in long-term simulations (Figs 3 and 4; S9 and S10 Figs). What explicitly might comprise a 50% reduction in NPIs? Empirically, this may result from measures that double an individual's average daily infective contacts; however, this is a complex question that requires significant modelling of social and spatial population dynamics that are beyond the scope of this current study. This model employs an oversimplification of NPIs into a single variable, whereas in reality NPIs are multifaceted and will dynamically change in composition over time as new resources, public health measures, and scientific knowledge change. Moreover, it is important to note that the effectiveness of NPIs not only relies on public health policy, but also the willingness for the public to embrace these measures.

Although this model employs data obtained from the SARS-CoV-2 outbreak in NYS, the conclusions made herein can be extrapolated to cities globally. To facilitate this extrapolation, the infectivity within the model is proportional to population density (ρ; S1 Table; see Meth-ods), as rapid spread of SARS-CoV-2 likely depends on high population densities [31]. Thus, geographically distinct regions can be compared through modulation of the population density variable. For example, while NYS experienced rapid SARS-CoV-2 transmission, the outbreak in the San Francisco Bay Area was significantly smaller. This was in part due to the earlier shel-ter in place order (March 16th, 2020 vs. March 22nd, 2020 in NYS), but may also reflect a less dense population [18, 32]. The San Francisco Bay area counties (San Francisco, San Mateo, Santa Clara, Alameda, and Marin) together have a population density of 1.809 (thousands per sq. mile), compared to the 2.726 of the New York counties in this study [33]. As a result, the San Francisco Bay Area would be predicted to have lower SARS-CoV-2 infectivity. This approach, however, is based on the assumption of a linear relationship between infectivity and population density. Instead a non-linear relationship may occur, in particular when comparing rural to urban locations. Thus, careful consideration must be employed when using this model in regions of relatively low population density. Future implementations of this model will be important to understand the differential transmission dynamics of SARS-CoV-2 in cities across the United States, and the world.

The conclusions of this study build upon the growing body SARS-CoV-2 modelling work by simulating the effects of NPIs, undocumented infection, seasonal infectivity, and sustained immunity in a single model of NYS transmission. Other studies have modelled these variables in part, and/or in other regions across the world. For example, a number of studies have con-cluded that NPIs are essential to mitigating SARS-CoV-2 transmission [13–17, 19]. Addition-ally, a recent study revealed that substantial undocumented infections were essential to the rapid dissemination of SARS-CoV-2 in China [5]. Seasonal infectivity has been analyzed by other models, which together also predict seasonal establishment of endemic SARS-CoV-2 transmission [14, 34]. Here, the effects of these variables were all incorporated into a single model; thereby enabling dynamic analysis of their effects on short- and long-term transmis-sion of SARS-CoV-2.

This model has a number of limitations. First, the model relies on data obtained from the NYSDOH, including public reports and press conferences. These data likely have errors in their estimations; thus, the model projections may overestimate or underestimate true epide-miological characteristics. Additionally, these data do not account for the relative lack in test-ing that occurred during the initial period of the NYS outbreak. As a result, the reported

confirmed cases likely are under representative, which may influence the model towards higher infectivity. Second, a fixed parameter approach was chosen for a number of variables including average incubation period ($\gamma^{-1}$), average time to hospitalization ($\delta^{-1}$), latency to non-hospitalized quarantine ($\theta^{-1}$), and the effective rate of undocumented infections (v; see S1 Appendix; S1 Table). The values of these parameters were chosen based on literature; however, in reality they likely change over time. Third, the seasonal variability SARS-CoV-2 transmission was modelled after non-SARS-CoV-2 HCoV seasonal variability. This assumption was conservatively weighted because the sensitivity of SARS-CoV-2 transmission to temperature and humidity is not yet known (see Methods); therefore, the true seasonal variability of SARS-CoV-2 may differ from the model's predictions. Lastly, pharmaceutical interventions, such as vaccines and therapeutic and prophylactic medications were not included in this model. It is unknown when these pharmaceutical interventions will be available, thus, there could be no confidence in any estimation of their emergence. However, these will likely represent the most critical long-term interventions in the global health arsenal to combat endemic SARS-CoV-2. The development of these pharmaceutical interventions will be essential to reduce the ultimate toll of this significant pandemic.

SARS-CoV-2 represents the most significant global health crisis of the 21st century. This modelling study simulates fundamental characteristics of this pathogen, the results of which can inform scientists and policymakers combating recurrent SARS-CoV-2 infection in the coming months to years. Simulations reveal that NPIs may be highly effective in reducing short-term morbidity and mortality, but only sustained immunity can likely prevent SARS-CoV-2 from emerging as an endemic pathogen. As a result, pharmaceutical interventions, such as highly effective drugs, antibodies, or vaccines will ultimately be necessary to combat SARS-CoV-2 in the future.

## Methods

### Data

SARS-CoV-2 case data for NYS were extracted from publicly reported clinical statistics by the New York State Department of Health, aggregated by the COVID Tracking Project [35, 36]. The data extracted were the total number of confirmed cases, total number of hospitalizations, active hospitalizations, and total number of deaths. Since the overwhelming majority of SARS-CoV-2 cases are concentrated in a limited number of NYS regions, only counties reporting greater than 1000 confirmed cases on (4/20/2020) were included (Queens, Kings, Nassau, Bronx, Suffolk, Westchester, Manhattan, Rockland, Richmond, Orange, Dutchess, Erie, and Monroe). To estimate population density of the affected NYS counties, population and land area estimates were obtained from the United States Census Bureau [33].

Non-SARS-CoV-2 human coronavirus (HCoV) viral testing data was obtained from the CDC National Respiratory and Enteric Virus Surveillance System (NRVESS) [37]. Data were extracted from two databases: viral testing data from 2014–2017 and 2018–2020 [38, 39]. To estimate the incidence of the HCoVs, the weekly percentage of positive tests were multiplied by the weekly population-weighted proportion of physician visits due to influenza-like illness (ILI) reported by the US Outpatient Influenza-like Illness Surveillance Network (ILINet) [38]. This method was based on methods published elsewhere [14, 40]. All data were compiled in Matlab (*Mathworks*) and prepared with custom-built software.

### Transmission model of SARS-CoV-2

The transmission dynamics of SARS-CoV-2 were modelled with a system of ordinary differential equations (ODEs) describing an eleven-compartment susceptible-exposed-infected-

recovered-susceptible (SEIRS) model (see S1 Appendix) [41]. The schematic of the model is shown in S1 Fig. "Confirmed active infections" were calculated as the sum of Q and H. The compartment P represents individuals protected by NPIs. The following coefficients were implemented as functions of time: $\alpha(t)$, $\zeta(t)$, $\lambda(t)$, $\kappa(t)$, and $\chi(t)$. To model the implementation of NPIs, $\alpha = 0$ before NPIs were in place, and became a fitted constant after (NPIs initiated in NYS on March 22nd, 2020) [18]. The "leak" of individuals from the protection of NPIs was modelled with $\zeta(t)$, where the magnitude of $\zeta$ was manipulated in simulations to represent the lessening of NPIs. The recovery and death rate of hospitalized patients were modelled with $\lambda(t)$ and $\kappa(t)$ respectively, which were composed of the follow one-phase exponential association and decay equations:

$$\lambda(t) = \lambda + \frac{\sigma}{\lambda}\left(1 - e^{-\epsilon t}\right)$$

$$\kappa(t) = \frac{\kappa}{\mu} + \left(\kappa - \frac{\kappa}{\mu}\right)\left(e^{-\psi t}\right)$$

Seasonal variability of SARS-CoV-2 infectiveness was modelled with $\chi(t)$. To do so, the weekly incidence of HCoVs from 2014–2020 were averaged to create a generalized annual model of HCoVs seasonal variation. It is yet unknown how the infectiveness COVID may vary with weather. Recent studies including retrospective analyses, *in vitro* experiments, and epidemiological models suggest there may be a component of weather-based variability; however, this may be significantly less other respiratory viruses [42–48]. Based on these studies, the seasonal variability of SARS-CoV-2 ($\chi(t)$) was generated by conservatively weighting the generalized annual model of HCoV seasonal variability by a factor of 0.2 (S5 Fig).

Undocumented infections represent an increasingly important factor in the spread of SARS-CoV-2. From serological studies, an undocumented rate of 75% was estimated by dividing the total number of confirmed cases by the product of the percent of serologically positive individuals and the population of the most affected NYS counties (see Methods, Data), subtracted from one [8–10]. To calculate $\nu$, the following formula was used:

$$\nu = \frac{Rate_{Undocumented}(\theta + \delta)}{1 - Rate_{Undocumented}}$$

The duration and annual recurrence of SARS-CoV-2 infections depends on the fraction of people who develop protective immunity to SARS-CoV-2, and the duration of immunity. It is currently unclear how individuals will respond to SARS-CoV-2 infection; however, data from similar HCoVs may provide an estimated range. Studies suggest that HCoV-OC43 and HCoV-HKU1 confer protective immunity that lasts less than a year, whereas immunity to SARS-CoV-2 may last much longer [24, 25]. To model immunity, the transition rate from "Recovered" compartments ($R_A$, $R_Q$, and $R_H$) to the "Susceptible" compartment depended on the following relationship:

$$Waning\ Immunity\ Rate = \frac{\pi}{1 - \tau}$$

Where $\pi$ represented the inverse of the duration of immunity, and $\tau$ represented the fraction of individuals who do not develop immunity.

Parameter estimation, sensitivity analysis, goodness of fit analysis, and reproductive number proofs are described in S1 Appendix.

## Simulations and data analysis

Simulations and output figures were generated in Matlab (*Mathworks*), with custom-built software. Figures were edited in Adobe Illustrator (*Adobe*).

## Supporting information

**S1 Appendix. Supplementary methods.**
(PDF)

**S1 Table. Fixed parameters.**
(PDF)

**S2 Table. Fit parameters.**
(PDF)

**S3 Table. Simulating the effect of undocumented infections on SARS-CoV-2 transmission, simulation results on September 1st, 2020, related to Fig 1.**
(PDF)

**S4 Table. Reduction of NPIs by >50% may result increased transmission and significant mortality, simulation results on September 1st, 2020, related to Fig 2.**
(PDF)

**S5 Table. Short-term effects of phased relaxed NPIs in NYS, simulation results on September 1st, 2020, related to Fig 2 and S8 Fig.**
(PDF)

**S1 Fig. Schematic of SARS-CoV-2 transmission model.** S (susceptible), E (exposed individuals), I (infected), U (undocumented), Q (quarantined), H (hospitalized), $R_U$ (recovered undocumented), $R_Q$ (recovered quarantined), $R_H$ (recovered hospitalized), D (dead), and P (protected).
(PNG)

**S2 Fig. Bivariate sensitivity analysis of model parameters with respect to total deaths, related to Fig 1.**
(PNG)

**S3 Fig. Global sensitivity analysis of model parameters with respect to total deaths, related to Fig 1.**
(PNG)

**S4 Fig. Goodness of fit analysis, related to Fig 1.**
(PNG)

**S5 Fig. Annualized model of SARS-CoV-2 seasonal variability, related to Fig 1.**
(PNG)

**S6 Fig. Simulating the effect of undocumented infections on SARS-CoV-2 transmission, comprehensive model outputs, related to Fig 1.**
(PNG)

**S7 Fig. Short-term effects of relaxed NPIs in NYS, comprehensive outputs, related to Fig 2.**
(PNG)

**S8 Fig. Short-term effects of phased relaxed NPIs in NYS, related to Fig 2.** (A-F). Simulation of SARS-CoV-2 transmission dynamics in the presence of NPIs through September 1,

2020. Periods of NPIs signified as in (A) top: pink, increased NPIs; green, relaxed NPIs. Orange circles, NYS SARS-CoV-2 data. Lines, simulated projection of reduced NPIs starting June 1, 2020. A. Active confirmed infections. B. Active hospitalizations. C. Cumulative deaths. D. Heatmap displaying the effect of NPI magnitude and date of reduction on the number of cumulative deaths. E. $R(t)$. F. Categorical heatmap displaying the effect of NPI magnitude and date of reduction on $R(t) > 1$ (yellow, $R(t) > 1$; blue $R(t) < 1$). (G-H). Simulation of extreme reduction of NPIs on June 1, 2020. G. Active hospitalizations. H. Cumulative deaths. (PNG)

**S9 Fig. Recurrent outbreak of SARS-CoV-2 in NYS in early 2021, comprehensive outputs, related to Fig 3.**
(PNG)

**S10 Fig. Simulation of sustained immunity to SARS-CoV-2 predicts endemic potential, comprehensive outputs, related to Fig 4.**
(PNG)

## Acknowledgments

Thank you to Dr. Stephen L. Hoffman, Dr. B. Kim Lee Sim, Dr. Seth A. Hoffman, Dr. Ellen A. Lumpkin, and Dr. Thomas L. Richie for helpful discussions and comments on the manuscript; Dr. Steven Shea for essential feedback on preliminary studies; and Mr. Abulhair Saparov for input on the methods and analysis.

## Author Contributions

**Conceptualization:** Benjamin U. Hoffman.

**Data curation:** Benjamin U. Hoffman.

**Formal analysis:** Benjamin U. Hoffman.

**Funding acquisition:** Benjamin U. Hoffman.

**Investigation:** Benjamin U. Hoffman.

**Methodology:** Benjamin U. Hoffman.

**Project administration:** Benjamin U. Hoffman.

**Resources:** Benjamin U. Hoffman.

**Software:** Benjamin U. Hoffman.

**Supervision:** Benjamin U. Hoffman.

**Validation:** Benjamin U. Hoffman.

**Visualization:** Benjamin U. Hoffman.

**Writing – original draft:** Benjamin U. Hoffman.

**Writing – review & editing:** Benjamin U. Hoffman.

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
