## [Decision Letter · Decision Letter 0]

23 Jul 2020

PONE-D-20-13695

Significant Relaxation of SARS-CoV-2-Targeted Non-Pharmaceutical Interventions Will Result in Profound Mortality: A New York State Modelling Study

PLOS ONE

Dear Dr. Hoffman,

Thank you for submitting your manuscript to PLOS ONE. After careful consideration, we feel that it has merit but does not fully meet PLOS ONE’s publication criteria as it currently stands. Therefore, we invite you to submit a revised version of the manuscript that addresses the points raised during the review process.

We look forward to receiving your revised manuscript.

Kind regards,

Lucy C. Okell

Academic Editor

PLOS ONE

Journal Requirements:

2. Please include captions for your Supporting Information files at the end of your manuscript, and update any in-text citations to match accordingly. Please see our Supporting Information guidelines for more information: http://journals.plos.org/plosone/s/supporting-information

Reviewers' comments:

Reviewer's Responses to Questions

**Comments to the Author**

1. Is the manuscript technically sound, and do the data support the conclusions?

Reviewer #1: Yes

Reviewer #2: Partly

2. Has the statistical analysis been performed appropriately and rigorously? 

Reviewer #1: Yes

Reviewer #2: N/A

3. Have the authors made all data underlying the findings in their manuscript fully available?

Reviewer #1: Yes

Reviewer #2: Yes

4. Is the manuscript presented in an intelligible fashion and written in standard English?

Reviewer #1: Yes

Reviewer #2: Yes

5. Review Comments to the Author

Reviewer #1: This is a high-quality modeling study examining the effects of social distancing relaxation, undocumented infection, seasonal infectivity and immunity on SARS-CoV-2 transmission in New York state. The study is well motivated, the analyses are technically sound, and the manuscript is clearly written. Here I have a few questions that I hope the author can clarify.

1. In model fitting, is the model fitted to the cumulative cases (confirmed cases, hospitalizations and deaths) or daily new cases? How are the variables in the equations (i.e., active cases, not new cases) mapped to observations?

2. “Serological data from NYS indicate that ~10-20% of the affected population have detectable antibodies to SARS-CoV-2, even though only ~2% have tested positive thus far.” Based on this statement, it seems the ascertainment rate is between 10% to 20%. Why a 75% undocumented rate (a 25% ascertainment rate) is used?

3. It would be good to present the seasonality function \\chi(t) estimated from HCoV.

4. Is the delay from infection acquisition to case confirmation considered in the model?

5. “The infectivity within the model is proportional to population density.” The infectivity and population density should be positively correlated. However, the exact form is complex. Maybe a proportional relationship is not the best choice. For instance, the population density in NYS could be 10 times of the population density in some rural areas; however, the infectivity in NYS may not be 10 times higher. For study in one particular state (like in this study), this assumption is fine as the scale of infectivity can be adjusted by the magnitude of \\beta. But claiming this proportional relationship seems too strong.

6. In the abstract, the peak basic reproductive number of 5.7 is conditioned on the many assumptions in the model. It should be stated explicitly.

7. On page 5, “SARS-CoV-2 outbreak will plateau”. I think it should be that the outbreak is declining and the cumulative cases plateau.

8. On page 10, a typo “in other regions across the word”.

Reviewer #2: This paper presents a modelling analysis on the New York COVID-19 epidemic and assesses the impact of reducing NPIs on controlling the epidemic.

Reading through the manuscript, it came across in certain sentences that modelling was uncovering/proving biological phenomena in COVID-19. For example "Undocumented infections drive SARS-CoV-2 transmission" and "Reduction of social distancing by >50% will result in dramatic mortality"

The role of mathematical modelling is not to 'prove' statements such as these, as our model projections are dependent on the assumptions WE have selected to include. For example, assuming values for the level of testing in NY, the relative infectiousness of asymptomatic, pre-symptomatic and symptomatic individuals and the relative duration of infectiousness, we are able to project the contribution incidence. So describing model projections should take the form of "assuming XYZ, the model projects that undocumented infections may be primary drivers of of SARS-CoV-2 transmission.

The paper states early on that the model is able 'capable of simulating the implementation of non-pharmaceutical interventions (NPIs) such as social distancing, contact tracing, and isolation' but then groups all interventions into one parameter alpha(t), a parameter that is fitted from the calibration process.

If a primary message from the paper is that reducing NPIs by more than 50% will result in dramatic mortality, then these NPIs should be modelled in more detail with their individual impact on model parameters captured. Otherwise the outcomes are not generalisable, and will themselves suffer from identifiability issues as it will be difficult to disentangle the impact of NPIs with the R0 and level of undocumented infection.

Additionally, a point should be included in the discussion, that a 50% reduction/ or any directed change in NPIs is not just a function of the policy enforced, but also public willingness. Lightly enforced restrictions from a government may be met with the a considerable public response resulting in more effective NPIs and vice versa.

The findings on seasonality and immunity should be rephrased to be hypothetical scenarios considered while global research is not yet conclusive on their impact.

The model findings in general should also be represented with uncertainty ranges.

6. PLOS authors have the option to publish the peer review history of their article (what does this mean?). If published, this will include your full peer review and any attached files.

Reviewer #1: **Yes: **Sen Pei

Reviewer #2: No

---

## [Author Response · Author response to Decision Letter 0]

24 Aug 2020

PONE-D-20-13695

Significant Relaxation of SARS-CoV-2-Targeted Non-Pharmaceutical Interventions Will Result in Profound Mortality: A New York State Modelling Study

PLOS ONE

Dear Dr. Hoffman,

Thank you for submitting your manuscript to PLOS ONE. After careful consideration, we feel that it has merit but does not fully meet PLOS ONE’s publication criteria as it currently stands. Therefore, we invite you to submit a revised version of the manuscript that addresses the points raised during the review process.

We look forward to receiving your revised manuscript.

Kind regards,

Lucy C. Okell

Academic Editor

PLOS ONE

Journal Requirements:

 I have reviewed all style requirements and to the best of my knowledge, the submitted manuscript meets them.

2. Please include captions for your Supporting Information files at the end of your manuscript, and update any in-text citations to match accordingly. Please see our Supporting Information guidelines for more information: http://journals.plos.org/plosone/s/supporting-information

I have reviewed all Supporting Information guidelines and to the best of my knowledge, the submitted manuscript meets all of them.

Reviewers' comments:

Reviewer's Responses to Questions

Comments to the Author

1. Is the manuscript technically sound, and do the data support the conclusions?

Reviewer #1: Yes

Reviewer #2: Partly

2. Has the statistical analysis been performed appropriately and rigorously? 

Reviewer #1: Yes

Reviewer #2: N/A

3. Have the authors made all data underlying the findings in their manuscript fully available?

Reviewer #1: Yes

Reviewer #2: Yes

4. Is the manuscript presented in an intelligible fashion and written in standard English?

Reviewer #1: Yes

Reviewer #2: Yes

5. Review Comments to the Author

Reviewer #1: This is a high-quality modeling study examining the effects of social distancing relaxation, undocumented infection, seasonal infectivity and immunity on SARS-CoV-2 transmission in New York state. The study is well motivated, the analyses are technically sound, and the manuscript is clearly written. Here I have a few questions that I hope the author can clarify.

1. In model fitting, is the model fitted to the cumulative cases (confirmed cases, hospitalizations and deaths) or daily new cases? How are the variables in the equations (i.e., active cases, not new cases) mapped to observations?

Response: Thank you for comment. I apologize that the methods were not as clear as they could have been in the first submission. The data extracted from the NYSDOH were as described by the following text in the Methods section:

“The data extracted were the total number of confirmed cases, total number of hospitalizations, active hospitalizations, and total number of deaths.” (Methods, Data, Page 15, Lines 687-698)

In order to fit the model to these data, the model compartments were extrapolated from the case data as described in the following updated text in the Supplementary Methods section (S1 Appendix):

“Model compartments Q, Rh, H, D were fit to extracted NYSDOH case data (confirmed, hospitalized total, hospitalized active, deaths) according to the following transformations: Q = (confirmed) – (hospitalized total); Rh = (hospitalized total) – (hospitalized active) – (deaths); H = (hospitalized active); D = (deaths).” (S1 Appendix, Supplementary Methods, Parameter Estimation)

In exploring the NYSDOH data, there is a discrepancy in the reported daily case data and the cumulative case data—thus, only cumulative data (with the exception of active hospitalizations) were used. The variables of the model were estimated as described by the follow text in the Results section:

“The model was optimized to SARS-CoV-2 case data from New York State collected between March 4th and April 28th, 2020 with the particle swarm algorithm, and confidence intervals were estimated by introducing lognormal gaussian noise to the source data and randomly sampling initial parameter estimates (see Methods; S1-2 Tables) [11, 12].” (Results, Page 4, Lines 132-135)

Additional variables in equations were estimated by solving the system of ordinary differential equations (described in Supplemental Methods), based the specific conditions of each simulation.

2. “Serological data from NYS indicate that ~10-20% of the affected population have detectable antibodies to SARS-CoV-2, even though only ~2% have tested positive thus far.” Based on this statement, it seems the ascertainment rate is between 10% to 20%. Why a 75% undocumented rate (a 25% ascertainment rate) is used?

Response: Thank you for this comment. Ascertainment bias is an important factor in the accuracy of epidemiological modelling, particularly in the early stages of epidemics, where testing and surveillance capacity is still growing. To estimate the undocumented infection rate, the following method was used:

“From serological studies, an undocumented rate of 75% was estimated by dividing the total number of confirmed cases by the product of the percent of serologically positive individuals and the population of the most affected NYS counties (see Methods, Data), subtracted from one” (Methods, Page 16, Lines 724-726)

This method explicitly estimates the fraction of documented cases amongst serologically identified cases, and as a result the fraction of undocumented cases. While other methods exist to estimate these values, given the limited serological data available at the time of writing this manuscript, this simplistic methodology was chosen. 

3. It would be good to present the seasonality function \\chi(t) estimated from HCoV.

Response: Thank you, I agree that data sharing and transparency of methodology are essential to constructive scientific discourse. Particularly during the current global crises of COVID-19, it is of paramount importance that researchers make available all data and resources they have used. I have updated the Supplemental Information to include a new figure (Figure S5) displaying the annualized seasonality function. In addition, all methodology and exact values used to estimate the seasonality function are freely available at the manuscript database below:

https://github.com/buh2003/BUHoffman_COVID

4. Is the delay from infection acquisition to case confirmation considered in the model?

Response: Thank you for this comment. The delay from infection acquisition to case confirmation is an important component of epidemiological modelling. The compartment model used in this study does incorporate an “Exposed” compartment (the E in the classical SEIR model), which is composed of individuals who are exposed to the virus, but not yet infectious. However, this model does not consider undocumented infectious individuals (U in the model) who later become documented through testing. Given the overall paucity of testing capacity at the time of writing this manuscript, not incorporating this transition should have limited impact on the model’s simulations. However, as testing capacity has greatly increased in more recent month, it will be essential to incorporate this transition in current and future studies. 

5. “The infectivity within the model is proportional to population density.” The infectivity and population density should be positively correlated. However, the exact form is complex. Maybe a proportional relationship is not the best choice. For instance, the population density in NYS could be 10 times of the population density in some rural areas; however, the infectivity in NYS may not be 10 times higher. For study in one particular state (like in this study), this assumption is fine as the scale of infectivity can be adjusted by the magnitude of \\beta. But claiming this proportional relationship seems too strong.

Response: Thank you for this comment. Incorporating a proportional relationship between infectivity and population density indeed is an oversimplified assumption. The goal of doing so was to contextualize the rate of daily infectious contacts in locations with differential population density. Likely, as you have stated, the relationship between population density and daily infectious contacts is likely non-linear. I have clarified this in the text as below:

“This approach, however, is based on the assumption of a linear relationship between infectivity and population density. Instead a non-linear relationship may occur, in particular when comparing rural to urban locations. Thus, careful consideration must be employed when using this model in regions of relatively low population density.” (Discussion, Page 12, Lines 622-626)

6. In the abstract, the peak basic reproductive number of 5.7 is conditioned on the many assumptions in the model. It should be stated explicitly.

I agree that the peak basic reproductive number is based on numerous assumptions. As a result, I no longer think it is a primary finding, and have removed it from the abstract. I have clarified this finding in the text as below:

“One important finding of this study is that based on the conditions of the model, simulations indicate that undocumented infections may have fueled the rapid spread of SARS CoV 2 in NYS. During the recent NYS outbreak, the model estimated a peak R0 of 5.7 (5.3-6.0, 95% CI), reaching a baseline of 4.4 (4.1-4.7, 95% CI) by July 2020, greater than reported in other publications (S6 Fig): which estimated 2.2 in the United States, and 2.2-2.6 in Wuhan and other regions of China [5, 13, 14, 27]. The magnitude of the model’s estimated R0 reflects the assumption that a large number of undocumented infections existed in NYS at the start of the outbreak. Based on serological studies, a 75% undocumented infection rate was implemented in this study (see Methods) [8-10]. As a result, the model projects 868,000 (795,000-947,000, 95% CI) undocumented infections by April 28th, 2020, despite the NYSDOH reporting only 295,106 confirmed infections. It is important to note, however, that estimation of R0 during the initial outbreak of an epidemic is confounded by the concurrent exponential increase in infections and testing. To account for this, estimation of the maximum basic reproductive number was approximated only when the percent positivity of confirmed tests was negative for 7 consecutive days (see S1 Appendix). However, despite this method, it is possible the findings of this study may represent overestimation of infectivity.” (Discussion, Pages 10-11, Lines 561-594)

7. On page 5, “SARS-CoV-2 outbreak will plateau”. I think it should be that the outbreak is declining and the cumulative cases plateau.

Thank you for this comment. I agree with your suggestion. I have changed the text as below:

“Based on these conditions, projecting through September 1st, 2020, if the NPIs implemented on March 22nd, 2020 are not relaxed, the SARS CoV 2 outbreak will likely be in decline by mid-July 2020, with a plateau in cumulative cases: 1.53 million people in NYS will be infected (undocumented infected+ confirmed symptomatic infected; 1.26-1.79 million, 95% CI), with 100,000 total hospitalizations (80,000-120,000, 95% CI), and 28,500 deaths (22,000-37,600, 95% CI; Fig 1D-F; S3 Table) [18].” (Results, Page 5, Lines 191-196)

8. On page 10, a typo “in other regions across the word”.

Response: Thank you for catching this typo, the text has been corrected.

Reviewer #2: This paper presents a modelling analysis on the New York COVID-19 epidemic and assesses the impact of reducing NPIs on controlling the epidemic.

Reading through the manuscript, it came across in certain sentences that modelling was uncovering/proving biological phenomena in COVID-19. For example "Undocumented infections drive SARS-CoV-2 transmission" and "Reduction of social distancing by >50% will result in dramatic mortality"

The role of mathematical modelling is not to 'prove' statements such as these, as our model projections are dependent on the assumptions WE have selected to include. For example, assuming values for the level of testing in NY, the relative infectiousness of asymptomatic, pre-symptomatic and symptomatic individuals and the relative duration of infectiousness, we are able to project the contribution incidence. So describing model projections should take the form of "assuming XYZ, the model projects that undocumented infections may be primary drivers of of SARS-CoV-2 transmission.

Response: I greatly appreciate this clarification of the appropriate language used to describe the model’s findings. I have adjusted the text throughout the manuscript to reflect the notion that the results are predicated on the assumptions and conditions of the model. Key examples of adjust text are as below:

Figure 1 title: Simulating the effect of undocumented infections on SARS-CoV-2 transmission

Figure 2 title: Reduction of NPIs by >50% may result increased transmission and significant mortality

Figure 4 title: Simulation of sustained immunity to SARS-CoV-2 predicts endemic potential

“Within the epidemiological conditions explored in this study, simulations indicate that SARS-CoV 2 is a highly infectious pathogen, whose transmission dynamics may be driven by large numbers of undocumented infections.” (Introduction, Page 3, Lines 93-95)

“These simulations suggest that the infectious potential of SARS CoV 2 has thus far been underestimated, and that undocumented infections may be primary drivers of transmission.” (Results, Page 5, Lines 202-204)

“Based on the conditions of these simulations, these data indicate that reducing NPIs by up to 50% in NYS may not significantly increase transmission before September 1st, 2020.” (Results, Page 6, Lines 248-250)

“Importantly, the increased winter-time transmission reflects the model’s assumption that the seasonal variation in transmission dynamics of SARS CoV 2 are similar to that of other non SARS CoV 2 human coronaviruses. Given this assumption, simulations predict recurrent outbreak in early 2021 that may be mitigated, but not avoided entirely, through the resumption of NPIs.” (Results, Page 8, Lines 463-467)

The paper states early on that the model is able 'capable of simulating the implementation of non-pharmaceutical interventions (NPIs) such as social distancing, contact tracing, and isolation' but then groups all interventions into one parameter alpha(t), a parameter that is fitted from the calibration process.

If a primary message from the paper is that reducing NPIs by more than 50% will result in dramatic mortality, then these NPIs should be modelled in more detail with their individual impact on model parameters captured. Otherwise the outcomes are not generalisable, and will themselves suffer from identifiability issues as it will be difficult to disentangle the impact of NPIs with the R0 and level of undocumented infection.

Response: Thank you for these important comments. I agree that this model employs a simplistic conceptualization of NPIs into a single variable. This limits the exploration of the differential importance specific types of NPIs and the combination of NPIs that x% reduction represents in reality. While I understand the importance of modelling NPIs in more detail, this would require significant reworking of the fundamental makeup of the model, and I unfortunately believe this is beyond the scope of this current manuscript. However, I believe this is an essential topic that should be explored in future work. I have clarified these points in the discussion as below:

“What explicitly might comprise a 50% reduction in NPIs? Empirically, this may result from measures that double an individual’s average daily infective contacts; however, this is a complex question that requires significant modelling of social and spatial population dynamics that are beyond the scope of this current study. This model employs an oversimplification of NPIs into a single variable, whereas in reality NPIs are multifaceted and will dynamically change in composition over time as new resources, public health measures, and scientific knowledge change.” (Discussion, Page 11, Lines 585-591)

Additionally, a point should be included in the discussion, that a 50% reduction/ or any directed change in NPIs is not just a function of the policy enforced, but also public willingness. Lightly enforced restrictions from a government may be met with the a considerable public response resulting in more effective NPIs and vice versa.

Response: Thank you for this suggestion. I have included the following text in the discussion:

“Moreover, it is important to note that the effectiveness of NPIs not only relies on public health policy, but also the willingness for the public to embrace these measures.” (Discussion, Pages 11-12, Lines 591-610)

The findings on seasonality and immunity should be rephrased to be hypothetical scenarios considered while global research is not yet conclusive on their impact.

Response: Thank you for this comment. I have adjusted the text referring to seasonality and immunity throughout, examples as below:

“Importantly, the increased winter-time transmission reflects the model’s assumption that the seasonal variation in transmission dynamics of SARS-CoV-2 are similar to that of other non-SARS-CoV-2 human coronaviruses. Given this assumption, simulations predict recurrent outbreak in early 2021 that may be mitigated, but not avoided entirely, through the resumption of NPIs.” (Results, Page 8, Lines 427-431)

“Indeed, given the hypothetical conditions of the simulations, SARS CoV 2 emerged as an endemic pathogen in NYS despite resumption of NPIs for 3 months in the winter of 2020/21.” (Results, Page 9, Lines 497-499)

The model findings in general should also be represented with uncertainty ranges.

Response: Thank you for pointing out this critical oversight. I have now included 95% CIs for all data cited in the text, and have included 3 new supplementary tables (Tables S3-5) which provide further results, including 95% CIs.

---

## [Editor Report · Decision Letter 1]

11 Sep 2020

Significant Relaxation of SARS-CoV-2-Targeted Non-Pharmaceutical Interventions May Result in Profound Mortality: A New York State Modelling Study

PONE-D-20-13695R1

Dear Dr. Hoffman,

We’re pleased to inform you that your manuscript has been judged scientifically suitable for publication and will be formally accepted for publication once it meets all outstanding technical requirements.

Kind regards,

Lucy C. Okell

Academic Editor

PLOS ONE

---

## [Editor Report · Acceptance letter]

16 Sep 2020

PONE-D-20-13695R1 

Significant Relaxation of SARS-CoV-2-Targeted Non-Pharmaceutical Interventions May Result in Profound Mortality: A New York State Modelling Study 

Dear Dr. Hoffman:

I'm pleased to inform you that your manuscript has been deemed suitable for publication in PLOS ONE. Congratulations! Your manuscript is now with our production department. 

Kind regards, 

on behalf of

Dr. Lucy C. Okell 

Academic Editor

PLOS ONE